# Green Financial Supervision Information System Based on Genetic Algorithm Optimization under Carbon Peaking and Carbon Neutrality Goals

**Wangfangyu Wan**

School of Mathematics, University of Edinburgh, Edinburgh EH9 3FD, Scotland, UK; wwfy888@126.com

**Abstract:** Under the guidance of "carbon peaking and carbon neutrality", "ecological priority" and "green development" have become the popular consensus, and the financial regulatory level continuously guides financial institutions to increase investment in green and low carbon projects. In the field of green financial supervision in China, due to imperfect systems and poor adaptability, financial risks are often difficult to control within a reasonable range, which has had a significant impact on financial supervision and management. This article aimed to optimize the green financial regulatory information system under the carbon peaking and carbon neutrality goals. Firstly, this article analyzed the concept and background of green finance regulation; then, an investigation was conducted on the construction of the green finance service information system, and a green finance information system supervision plan was established. Finally, data collection and analysis were conducted, and the supervision of the green finance information system was carried out using a standard genetic algorithm based on a fuzzy evaluation matrix. This article used a genetic algorithm to optimize the green financial regulatory information system, and selected 500 people to evaluate the use of the system before and after the optimization. The proportion of very satisfied people increased from 11.2% to 19.2%; the proportion of satisfied people increased from 17.2% to 37.6%; the proportion of people who were very dissatisfied decreased from 14.4% to 3.6%. The experiment in this article showed that the optimized system could operate more stably, and the process was more reasonable. The statistical analysis ability was significantly enhanced, and the functions were more comprehensive. This suggests that the system could better regulate the development of green finance.

**Keywords:** carbon peaking; carbon neutrality; green finance; regulatory information system; genetic algorithm

## 1. Introduction

Against the backdrop of global climate change, environmental protection, and sustainable development, green finance (abbreviated as GF for convenience) has become an important industry of widespread concern to the international community [1]. The current green finance market has problems such as information asymmetry, lack of standards and regulations, and insufficient supervision, which seriously restrict the development and regulation of the green finance market and also increase market risks and uncertainties. The green finance regulatory information system can effectively achieve intelligent regulation of the green finance market, but the current system's data analysis capabilities have certain limitations, making it difficult to provide more reliable data analysis and decision-making support for green finance regulation. With the development of mathematical optimization theory, genetic algorithms (GAs) have made great progress and have been widely applied in various professional fields. In the construction of a green finance regulatory information system, multiple factors and constraints need to be considered. GAs can search for multiple candidate solutions in parallel, thereby finding the global optimal solution and providing efficient, accurate, and interpretable green finance regulatory optimization decisions.

This has important practical value for promoting the sustainable development of green finance regulation.

Against the backdrop of "carbon peak and carbon neutrality", green finance regulation is of great significance to the Chinese economy. It provides reliable data support and a basic guarantee of support for the construction of a top-level system of GF and the promotion of the development of GF. However, there are still several problems in the implementation practice [2]. Ma Jason Z's research examined the impact of corporate social responsibility information disclosure on green investment and analyzed whether this impact was different during the Chinese stock market crash. The research results revealed the mismatch between green investment and corporate social responsibility disclosure, and that market regulation played a regulatory role [3]. Park Hyoungkun believed that green banking created new business opportunities for private sector banks and expanded the task of regulators to manage individual financial institution (abbreviated as FI for convenience) risks [4]. Hsu Ching-chi explored the relative role of green technology innovation in promoting the development of GF in the western and central regions. He believed that green financing reduced short-term loans, thereby limiting excessive investment in clean energy. At the same time, GF growth would improve investment productivity in renewable energy [5]. Debrah Caleb conducted a systematic review of the current status and trends in green finance research using a hybrid approach of bibliometrics and qualitative analysis. A bibliometric review was conducted on 995 relevant publications retrieved from Scopus, and validation was conducted through Web of Science, Google Scholar, and ScienceDirect. The author has a deep understanding of the key applications of green finance in specific research fields and the ways to achieve green finance regulation [6]. In order to create conditions for ensuring a fair competitive environment between traditional and green economies, Falcone Pasquale Marcello conducted in-depth research on green finance regulation and concluded that the establishment of regulatory systems requires a sustainable transition based on deeply rooted production and consumption patterns [7]. The current research on green finance regulation has made good progress, but the financial data system is huge and complex, making it difficult for current regulatory methods to quickly and effectively achieve monitoring decision-making and analysis.

GAs have been applied to economic research, thus allowing for an understanding of economic issues at a deeper and broader level. Asima Mehdi believed that the application of GAs and hybrid models in financial markets could improve predictability and returns compared to other research models [8]. Ramachandran R believed that optimization was an important tool for decision-making and analyzing physical systems [9]. The financial services industry has recently witnessed new technological innovations and process disruptions. The entire industry and many financial technology startups are finding new ways to succeed in terms of new business models and service transformation methods. Industry and academic observers believe that this is not so much a series of minor changes as a revolution because financial services as a whole have made significant improvements in efficiency, customer focus, and information [10]. Considering that financial experts vent government spending on green recovery plans, this provides green banking risk openness. The components and methods of some green funds are differentiated, but they are very valuable for commercial banks and private enterprises [11]. China is in an era of major financial changes, and financial regulatory reform has received much attention. The financial industry is also showing a vigorous development trend under the influence of the country and the market. Starting from the theory and practice of financial regulation, the author first elaborated on the research background and global perspective of financial regulation under mixed operation, and then analyzed the current situation of financial market subject supervision under mixed operation in China. The current situation and problems faced by China's financial supervision (abbreviated as FS for convenience) have been summarized, and the excellent experience of several developed countries' FS systems has been analyzed and used for reference [12]. Based on China's basic national conditions, a sound FS has been proposed [13]. Policymakers and researchers have recently focused

on GF. Existing research on GF in the context of the banking sector has been reviewed. It is found that green securities, green investment, climate finance, carbon finance, green insurance, green credit, and green infrastructure bonds are part of the key green financial products of banks [14,15]. Banks are encouraged to participate in providing green financing as part of efforts to promote environmentally friendly economic activities for sustainable economic development [16]. Low carbon (LC) investment is currently concentrated in countries and regions with higher incomes, while risk mitigation activities are currently concentrated in countries and regions with lower incomes [17]. The main tool of GF in the field of decarbonization is green bonds; other types of GF are used to a limited extent [18,19]. GF can protect the environment, and develop the economy, so as to achieve sustainable development. Yi Xu proposed a genetic algorithm model based on adaptive improvement. The optimal solution to complex problems was obtained through linear adaptive optimization. The paper constructed an optimized comprehensive risk warning model and achieved automatic integration and prediction of financial risks. The experimental results show that the overall effectiveness of the algorithm in predicting financial risks improved, providing effective measures for achieving financial regulation [20]. Genetic algorithms can effectively improve the efficiency of green finance regulatory data analysis, but the current research on green finance regulatory information systems based on genetic algorithms makes it difficult to effectively meet the needs of different user groups in practical applications.

In order to improve the efficiency of green finance supervision and meet the needs of different groups for green finance supervision information systems, this article conducts in-depth research on the construction of green finance supervision information systems under the carbon peak and carbon neutrality goals, combined with genetic algorithm optimization. This article first analyzes the concept and background of green finance supervision, then investigates the construction of a green finance service information system, establishes a green finance supervision information system plan, and finally collects and analyzes data. The regulation of the green finance regulatory information system adopts a standard genetic algorithm based on a fuzzy evaluation matrix. To validate the green finance regulatory information system optimized based on a genetic algorithm, this article selected 500 people to evaluate the system usage before and after optimization and conducted an experimental analysis on them. The experimental results show that the green financial regulatory information system optimized based on a genetic algorithm has higher user satisfaction. In practical applications, green finance regulatory information systems optimized based on genetic algorithms can help improve the scientific analysis of green finance data and effectively promote the high-quality development of green finance regulation.

## 2. Necessity of Optimizing the Green Financial Regulatory Information System

GF regulatory data statistics aim to guide commercial banks to improve their GF capabilities and establish an efficient GF evaluation system, so as to enhance commercial banks' support for the implementation of proactive and responsible new development ideas. They provide a solid basis for policy formulation and promote economic green transformation and ecological civilization construction. However, there are limitations in the statistics of GF regulatory data. Therefore, GF regulatory data statistics need to be optimized, and the optimization process is shown in Figure 1.

Figure 1 was prepared by the author. As shown in Figure 1, to optimize the statistics of green financial regulatory data, it is necessary to first clarify the objectives, continuously improve the quality and efficiency of statistics, and expand the statistical caliber of financial products; the logical hierarchy of green attribute evaluation is divided and the evaluation process is optimized; supervision over the financing of pollution projects should be strengthened; data flow needs to be continuously optimized and data quality needs to be improved.

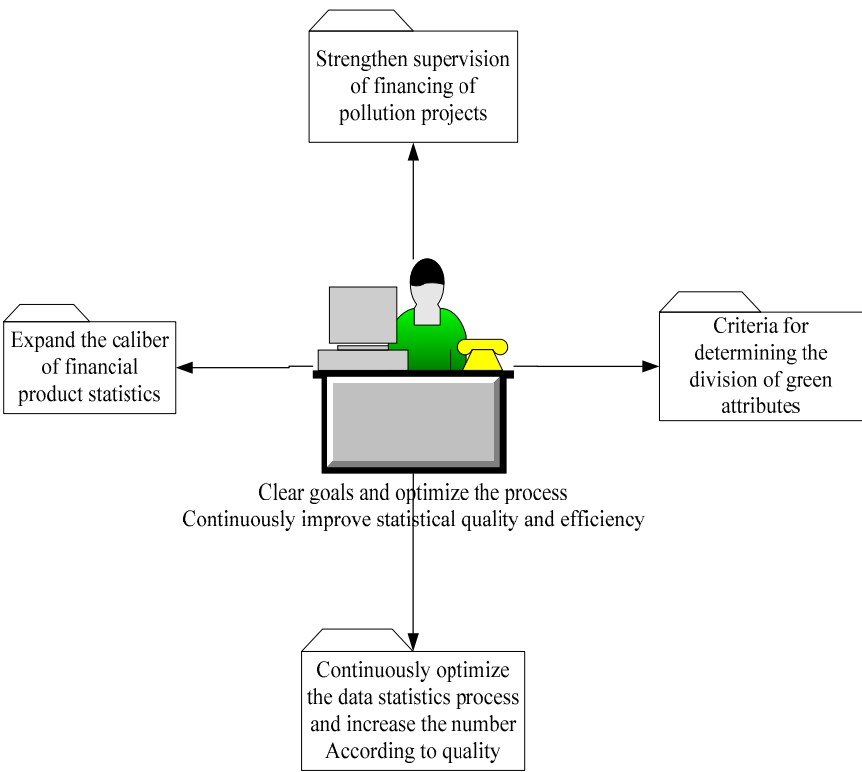

**Figure 1.** Optimization process of GF regulatory data.

## 2.1. Investigation of the Construction of a Green FS Information System

The financial ecosystem includes financial entities and the financial ecological environment [21]. Financial entities refer to FIs, financial markets, entities that can directly provide financial products and services to the people, and governments, enterprises, and individuals that consume financial products and services. Also included are staff whose main responsibilities include establishing norms, formulating policies, implementing supervision, and conducting regulation. They can play a crucial role in financial markets. A dynamic, balanced system of interaction, interdependence, and mutual influence is formed between various financial activity entities and the financial ecological environment on which they rely.

The green financial system consists of regulation, markets, institutions, and products. Regulation is the key [22]. Therefore, it is necessary to strengthen communication and collaboration among financial management departments, FIs, and relevant departments. A sustainable development strategy is reflected in financial operations. The implementation of the concept of GF requires a comprehensive discussion of ecological and environmental protection under the current financial operation and regulatory framework, rather than leaving finance.

With the widespread application of social informatization, currently, more and more applications are related to big data. On this basis, the collection, analysis, and processing of a large amount of data, as well as the extraction of useful information, are of great significance for China's green financial management. Further, valuable information is mined to assist the financial regulatory structure in timely grasping the dynamics of the market, thus formulating appropriate regulatory policies and improving the market regulatory system. A financial regulatory information-sharing mechanism has been established to build a shared database to improve regulatory efficiency. Figure 2 shows the construction scheme of the green FS information system.

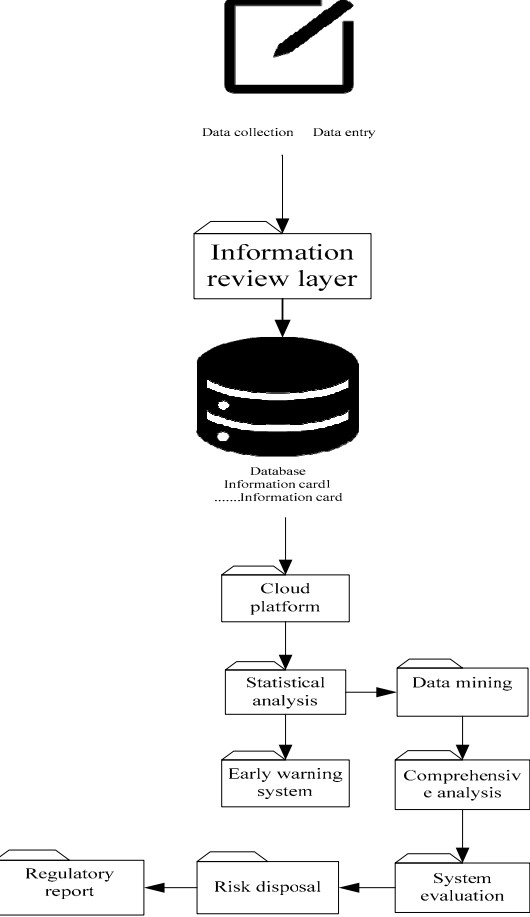

**Figure 2.** Construction scheme of a green FS information system.

Figure 2 was prepared by the author. In Figure 2, a green FS information system has been constructed, requiring data collection and input. The audited data are saved in the system database to achieve supervision, and daily statistical analysis of the process is performed to calculate various risks and generate various statistical reports. Useful information is grasped, and predictions and early warnings are made.

The outline of several emerging debates surrounding algorithm regulation highlights the legitimacy of algorithm regulation using insights from regulatory governance research, legal criticism, regulatory research, and key data research [23].

*2.2. Optimized Green FS Information System Based on a GA*

A GA is an evolutionary process that mimics "natural selection" and a "genetic mechanism" that is the result of evolution. Evolutionary laws in nature are simulated to find optimal solutions. Its basic characteristic is to directly operate on the constructed object, which has no restriction on derivation and no continuity of functions; the essence of this algorithm is implicit parallelism, with strong global optimization performance; the algorithm utilizes a probabilistic optimization algorithm to achieve automatic generation and guidance of the search space under unconstrained conditions and to achieve adaptive adjustment of the search direction. Due to the above characteristics, it has good application prospects in many fields. It is an extremely important information technology. GAs have strong systemic robustness. This is a characteristic robustness when the size and structure of the controlled object change. This is the key to the survival of the system under abnormal conditions [24]. Genetic algorithms can be used to optimize the decision-making and investment portfolio of green finance. The search and optimization process of genetic algorithms can help determine the best allocation of green investment portfolios to

maximize environmental and economic benefits. Financial innovation can provide new products, tools, and models for green finance. For example, innovative green financial products such as green bonds and carbon trading can attract investors to support and provide financing for green projects. Financial innovation can motivate and promote the wider financial community to participate in green economic development. Financial regulatory agencies can formulate and implement relevant policies and regulations to ensure the transparency, compliance, and sustainability of green financial activities. These regulatory measures help prevent risks in the field of green finance, protect the interests of investors, and provide a stable and reliable environment for green finance.

A new evaluation model was used. This model is built on the basis of a fuzzy evaluation matrix and uses a standard GA to check and modify, so as to calculate the weight of each element and use it in the green financial regulatory information system.

This paper transforms the optimization problem of the green finance regulatory information system into a mathematical model and defines a fitness function to evaluate the quality of each candidate solution. The fitness function is determined based on the optimization objective of maximizing system performance. The randomly generated set of initial candidate solutions is called a population. Each candidate solution is a configuration of the green financial regulatory information system. Candidate solutions in the population can be evaluated and ranked based on fitness functions, and individuals with higher fitness can be selected as parents. This article selects two individuals from the parents and generates new offspring individuals through cross-operation. It combines certain features of the parent individuals to generate a new individual. Mutation operations can be performed on offspring individuals to increase population diversity. Mutation operations can introduce new genes by randomly changing certain characteristics of individuals. Then, this article merges the parent and offspring individuals to form a new population. The experiment can repeat selection, crossover, and mutation operations, that is, repeat steps 3 to 6 until a solution that meets the optimization objectives is achieved. Finally, based on the evaluation results of the fitness function, the individual with the highest fitness is selected as the optimal solution, which is the optimal configuration of the green financial regulatory information system. Through the iterative process of the genetic algorithm, individuals in the population can be continuously optimized to gradually find better solutions, thus achieving the optimization of the green financial regulatory information system.

A new evaluation matrix is constructed using the consistency of evaluation indicators in the sample database to remove factors. A systematic, representative, and adaptive fuzzy comprehensive evaluation index system is established, and its relative membership fuzzy evaluation matrix is established using the sample data of each evaluation index [25]. To overcome the dimensional effect of various evaluation indicators and enhance the universality of evaluation methods, the sample data are standardized. The following standardized expression was developed:

The larger the better

$$r(i,j) = x(i,j)/[x_{max}(i) + x_{min}(i)] \tag{1}$$

The smaller the better

$$r(i,j) = [x_{max}(i) + x_{min}(i) - x(i,j)]/[x_{max}(i) + x_{min}(i)] \tag{2}$$

The more intermediate the better

$$r = (i,j)\begin{cases} x(i,j)/[x_{mid}(i) + x_{min}(i)], x_{min}(i) \leq x(i,j) < x_{mid}(i) \\ [x_{max}(i) + x_{mid}(i) - x(i,j)]/[x_{max}(i) + x_{mid}(i)], x_{mid}(i) \leq x(i,j) < x_{max}(i) \end{cases} \tag{3}$$

In the formula, $x_{max}(i)$, $x_{mid}(i)$, and $x_{min}(i)$, respectively, represent the maximum, median, and minimum values of the ith index in the scheme. $r(i,j)$ represents the standardized evaluation index value, which is equal to the relative membership value of the

ith evaluation index of the j scheme. $r = 1 \rightarrow n, j = 1 \rightarrow m$. $r(i, j)$ is selected as a unit, and a single indicator fuzzy evaluation matrix $R = [r(i, j)]_{nm}$ can be constructed.

In the above steps, based on the constructed fuzzy evaluation matrix $R = [r(i, j)]_{nm}$, a judgment matrix $B = (b_{ik})_{nm}$ is constructed to determine the weight of each evaluation index. The essence of a fuzzy comprehensive evaluation is an optimization process. If the change degree of the sample sequence $\{r(i_1, j) | j = 1 \rightarrow m\}$ of evaluation index i1 is greater than that of evaluation index i2, evaluation index i1 conveys more information than evaluation index i2. In order to avoid information asymmetry, this article introduces the sample standard deviations $s(i) = \left\{ \sum_{j=1}^{m} (r(i, j) - \overline{ri})^2 / m \right\}^{1/2}$ and i = 1→n of each indicator to reflect the role of each indicator in the comprehensive evaluation and to construct a judgment matrix B. In the formula, $\overline{ri} = \sum_{j=1}^{m} r(i, j) / m$ is used to represent the sample average value of each evaluation index, so the judgment moment can be constructed by using Formula (4).

$$b_{ik} \begin{cases} [s(i) - s(k)](b_m - 1)/(s_{max} - s_{min}) + 1 & s(k) \leq s(i) \\ 1/\{[s(i) - s(k)](b_m - 1)/(s_{max} - s_{min}) + 1\} & s(i) < s(k) \end{cases} \tag{4}$$

In Formula (4), $s_{max}$ and $s_{min}$ represent the maximum and minimum values of the $\{s(i) | i = i \rightarrow n\}$ pairs, respectively. In addition, the parameter value of the relative importance degree can be expressed as $b_m = \min\{9, \text{int}[s_{max}s_{min} + 0.5]\}$. Among these, int and min are represented as the respective rounding functions.

The judgment matrix B is subjected to consistency testing and correction, and the weight $w_i(i = 1, 2, \ldots, n)$ of each evaluation index is calculated. Among them, $w_i > 0$ and $\sum_{i=1}^{n} w_i = 1$. According to the definition of judgment matrix B, the formula is as follows [26]:

$$B_{ik} = w_i / w_k$$

$$(i = 1, 2, \ldots, n; k = 1, 2, \ldots n) \tag{5}$$

Matrix B has the following properties:

(1)  $b_{ii} = w_i / w_i = 1$; this represents the identity of the judgment matrix.
(2)  $B_{ki} = w_k / w_i = 1 / b_{ik}$; reciprocal property of judgment matrix.
(3)  $B_{ik}b_{kl} = (w_i / w_k)(w_k / w_1) = w_i w_1 = b_{il}$; consistency conditions of judgment matrices.

From the above properties, it can be shown that their interrelationships can be quantitatively transferred, and property (3) is a sufficient condition for property (1) and property (2).

The problem to be solved is to determine the weight value (WV) $w_i(i = 1, 2, \ldots, n)$ of each evaluation index from the known judgment matrix $B = (b_{ik})_{nm}$. If the judgment matrix satisfies Formula (5), the decision-maker can accurately measure $b_{ik} = w_i / w_k$. At this point, the judgment matrix B has perfect consistency, and the formula is obtained as follows [27]:

$$\sum_{i=1}^{n} \sum_{k=1}^{n} |b_{ik}w_k - w_i| = 0 \tag{6}$$

Due to design evaluation and human cognition, in practice, it is objective to find that the consistency conditions of judgment matrix B cannot be fully satisfied. If the consistency of B is not satisfactory, it must be modified. A modified decision matrix for B is recorded as $Y = \{y_{ik}\}_{nn}$, and the WV of each element representing it is also recorded as $w_i(i = 1, 2, \ldots, n)$. That is to say, the minimum matrix Y in Formula (7) is called the best consistent decision moment of B.

$$\text{minCIC}(n) = \sum_{i=1}^{n} \sum_{k=1}^{n} |y_{ik} - b_{ik}| / n^2 + \sum_{i=1}^{n} \sum_{k=1}^{n} |y_{ik}w_k - w_i| / n^2 \tag{7}$$

$$S.t. y_{ii} = 1 (i = 1, 2, \ldots, n)$$

$$1/y_{ki} = y_{ik} \in |b_{ik} - db_{ik} + db_{ik}| (i = 1, 2, \ldots, n; k = i + 1, \ldots, n)$$

$$W_i > 0 (i = 1, 2 \ldots, n) \tag{8}$$

$$\sum_{i=1}^{n} w_i = 1$$

Among these, Formula (7) is a function of the objective; CIC(n) represents the consistency index coefficient, and d is referred to as a non-negative parameter.

Obviously, Formula (7) belongs to a nonlinear optimization problem that is difficult to deal with using traditional methods. In the formula, the upper triangular matrix elements of the WV $w_i(i = 1, 2, \ldots, n)$ and the correction judgment matrix $Y = \{y_{ik}\}_{nn}$ are optimization variables. There are $n(n + 1)/2$ mutually independent optimization variables relative to the n-order judgment matrix B. It can be seen that the smaller the value CIC(n) on the left side of Formula (7), the higher the consistency of the judgment matrix B. If the global minimum value CIC(n) = 0 is taken, Y = B, and both Formula (7) and Formula (6) are valid. At this point, the judgment matrix B satisfies complete consistency, and it can be seen from the constraint condition Formula (8) that this global minimum value is unique. From the previous description, it can be seen that a GA is a general global optimization method for solving Formula (7)-type problems, and using it to solve Formula (7)-type problems is simple and effective. If the consistency index coefficient satisfies CIC(n) < 0.1, the judgment matrix would be considered to have satisfactory consistency. The WV $w_i(i = 1, 2, \ldots, n)$ of each evaluation index obtained through this method is acceptable; if this is not performed, the parameter d increases until the judgment matrix reaches a satisfactory consistency.

*2.3. Optimization of the Green Financial Supervision Information System*

On the basis of a genetic algorithm, this article optimizes the green finance regulatory information system in terms of three aspects: data collection, data analysis, and regulation. The data collection module is responsible for collecting relevant data on the green financial market, including information on green financial products, investors, issuers, rating agencies, etc.

The data analysis module utilizes data mining technology to analyze the collected data and explore the development trends, risks, and other information of the green financial market.

The regulatory module regulates the green financial market based on data analysis results to ensure fairness, impartiality, and stability in the market.

The core of this article lies in the application of genetic algorithms. A genetic algorithm is an optimization algorithm based on the theory of evolution. By simulating the process of biological evolution, starting from an initial population, it is gradually optimized to obtain the optimal solution. This paper uses genetic algorithms to optimize the green financial supervision information system, which mainly includes the following three aspects:

The optimization of the data collection module is achieved by optimizing the sampling strategy of the data collection module through genetic algorithms, thereby improving the efficiency and accuracy of data collection.

The optimization of the data analysis module optimizes the algorithms and parameters of the data analysis module through genetic algorithms, improving the efficiency and accuracy of data analysis.

Optimizing the regulatory module is achieved by optimizing the decision-making algorithms and parameters of the regulatory module through genetic algorithms, thereby improving the efficiency and accuracy of regulation.

### 3. Optimization of the Green FS Information System

There are many problems in the optimization of the GF regulatory information system. In order to solve the challenges and find solutions, this paper investigated the carbon emission data of China in recent years. The data in this article are sourced from carbon emission data officially released by the National Bureau of Statistics, the China Energy Administration, and the Ministry of Environmental Protection of China. Due to the incomplete publication and organization of carbon emission data from 2018 to 2022, and considering the impact of short-term factors such as economic fluctuations and policy changes in recent years, the data are unstable. To ensure the reliability of the experimental results in this article, a comprehensive analysis was only conducted on the data from 2007 to 2016. The data from 2007 to 2016 cover a relatively long period of time, allowing for better observation and evaluation of long-term trends and changes in carbon emissions.

(1)　The Development of GF

"Region B" is the first region in China to achieve the "two type" development goal, and has been widely referred in China. Therefore, research was conducted on LC development and GF in Region B, and a set of evaluation criteria systems for LC development and GF suitable for this region was constructed. In this way, repetitive research in urban agglomerations in other regions could be avoided and research resources could be saved, as shown in Figure 3 (including GDP (gross domestic product) and tce (ton of standard coal equivalent)):

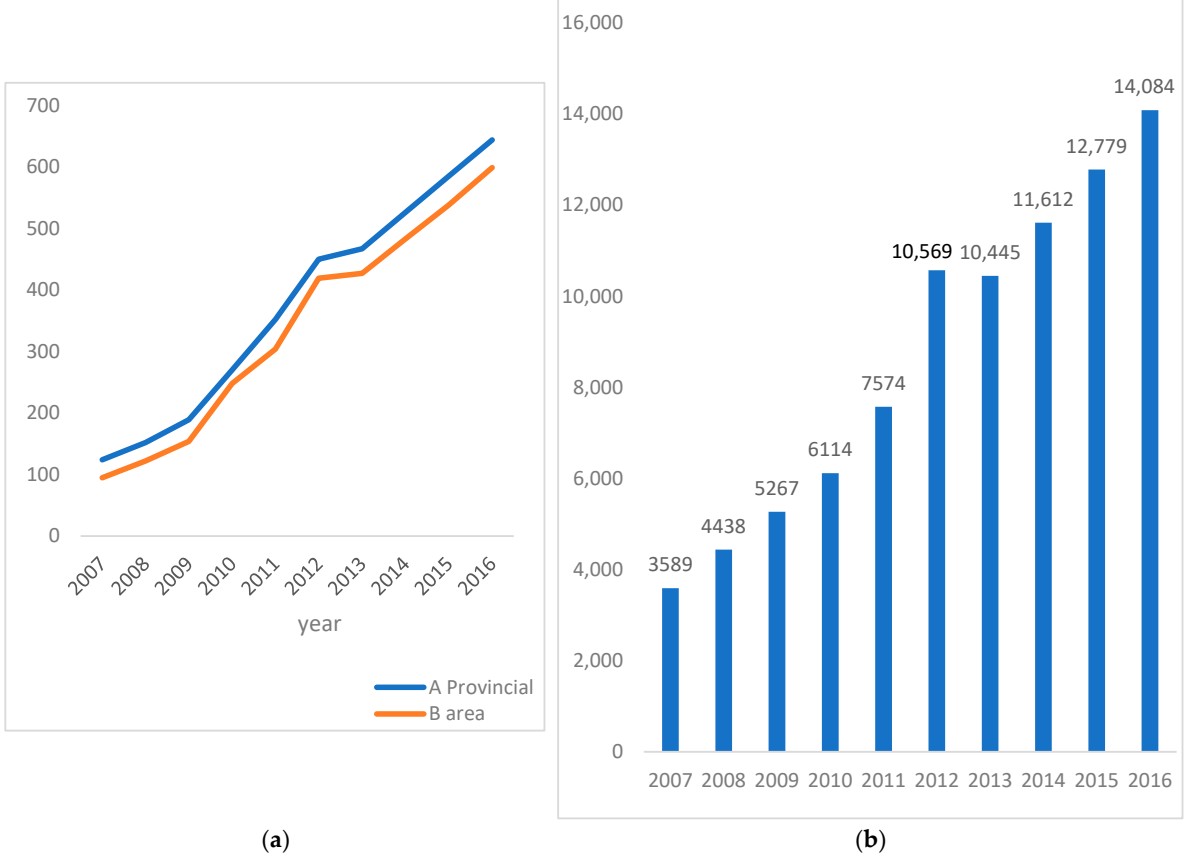

(**a**)　　　　　　　　　　　　　　　　　(**b**)

**Figure 3.** Carbon emissions growth trend. (**a**) GDP of carbon emissions in region B (unit: 10,000 CNY/tce); (**b**) China's carbon emissions GDP (unit: 10,000 CNY/tce).

Figure 3 was prepared by the author. As shown in Figure 3, the GDP per unit of carbon emissions of urban agglomerations in Region B increased year by year starting in 2007 and accounted for a large proportion in Province A. Compared with the national GDP per unit carbon emissions data, the GDP per unit carbon emissions in Region B also showed a trend of continuous growth. Its share of the country's total carbon emissions per unit of GDP also increased overall, from 2.65% in 2007 to 4.25% in 2016. This fully demonstrated that from the perspective of Province A, each carbon emission per unit of GDP of Region B would largely reflect the economic development trend of Province A. Therefore, in Province A, the economic development of Region B had important practical significance.

According to the results in Figure 3, this article analyzes the carbon emissions GDP growth data of Region B and the whole country using the green finance regulatory information system and verifies the accuracy of the system data analysis. The final results are shown in Table 1.

**Table 1.** Accuracy of system data analysis.

| Years | Accuracy of Data Analysis in Region B (%) | Accuracy of National Data Analysis (%) |
|---|---|---|
| 2007 | 90.21 | 93.05 |
| 2008 | 93.05 | 96.14 |
| 2009 | 91.17 | 91.12 |
| 2010 | 90.24 | 92.06 |
| 2011 | 93.05 | 91.53 |
| 2012 | 91.12 | 90.41 |
| 2013 | 90.11 | 93.32 |
| 2014 | 91.25 | 94.18 |
| 2015 | 91.38 | 90.02 |
| 2016 | 90.42 | 90.33 |

According to Table 1, it can be seen that in the analysis of carbon emissions GDP growth trend data, the accuracy of data analysis based on genetic algorithm optimization has shown good performance, with an accuracy rate of over 90% for annual data analysis.

(2)     The development status of GF in China

The country has actively developed GF. In this process, major FIs have launched corresponding green financial products, thus providing diversified financial services for the development of "carbon peaking and carbon neutrality", as shown in Figure 4.

Figure 4 was prepared by the author. As can be seen from Figure 4, China's carbon emissions increased several times from 2007 to 2016. Along with economic development, China's energy consumption was also significantly reduced, which greatly alleviated the environmental pollution problem. In 2016, the balance of various deposits of FIs was nearly four times that of 2007. In 2016, the balance of various loans of FIs was more than four times that of 2007. This figure showed that the contribution of each unit of carbon emissions to its gross domestic product was basically consistent with the growth of bank deposits and credit. This also indicated that China's FIs were more supportive of LC development, while China's GF showed a good development trend.

The advancement of an LC economy must be supported and assisted in various ways. Among these methods, the most important was financial support. How to achieve financial support for an LC economy was particularly important. The development of LC finance had important practical significance for promoting the advancement of LC technology and LC industry.

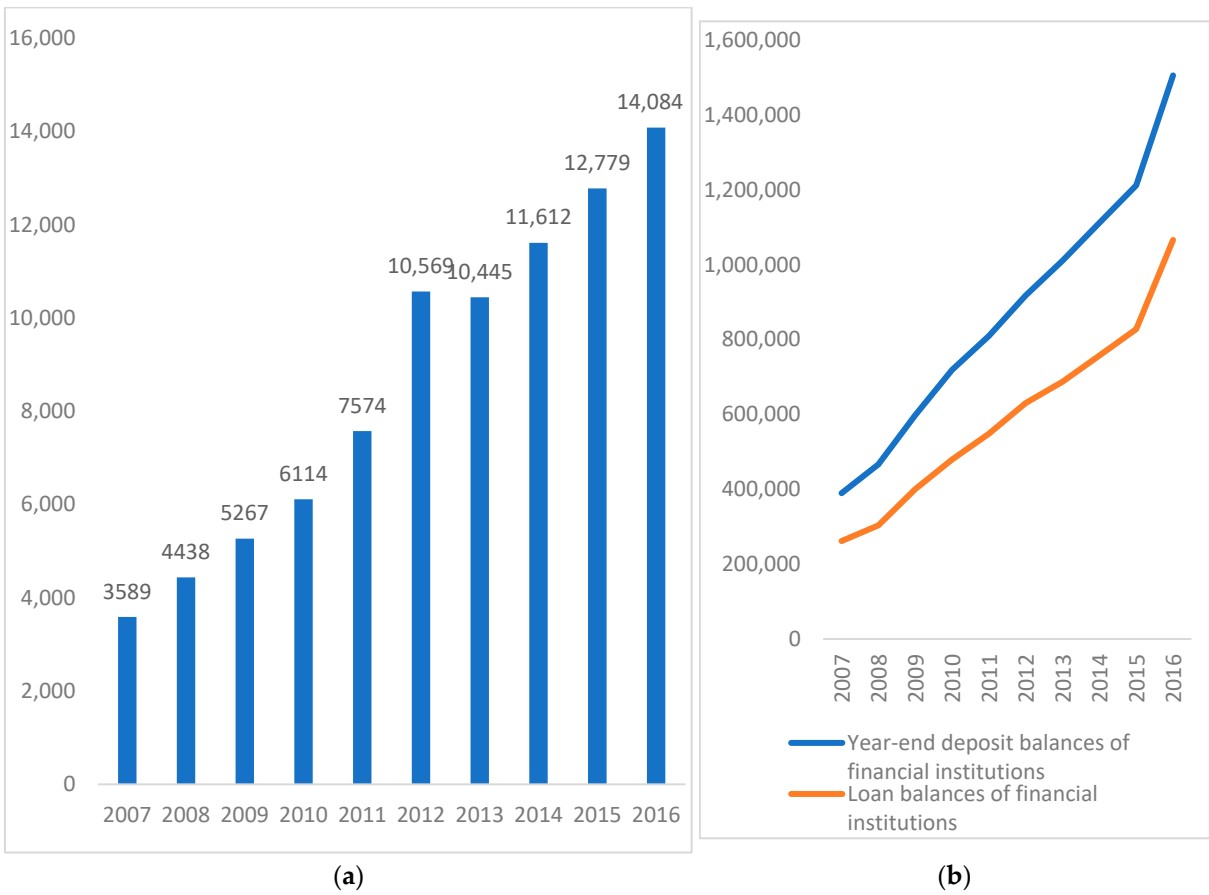

(**a**)                                       (**b**)

**Figure 4.** Balance of deposits and loans of FIs. (**a**) China's carbon emission GDP (unit: 10,000 CNY/tce); (**b**) Balance of deposits and loans of various FIs in China (unit/100 million CNY).

This article is also based on the data in Table 2 and uses a green finance regulatory system based on a genetic optimization algorithm to analyze the data. The final accuracy results are shown in Table 2:

**Table 2.** Accuracy of data analysis for green financial products.

| Years | Accuracy of China's Carbon Emissions GDP Data Analysis (%) | Analysis Accuracy of Deposit and Loan Balance Data (%) |
|---|---|---|
| 2007 | 93.52 | 90.11 |
| 2008 | 93.03 | 94.25 |
| 2009 | 96.15 | 91.07 |
| 2010 | 90.02 | 92.88 |
| 2011 | 91.32 | 90.16 |
| 2012 | 93.35 | 93.85 |
| 2013 | 90.14 | 92.02 |
| 2014 | 90.66 | 97.14 |
| 2015 | 95.72 | 90.33 |
| 2016 | 92.13 | 91.12 |

From Table 2, it can be seen that for the analysis of green finance product data over the years, the accuracy of the green finance regulatory information system optimized based on a genetic algorithm has reached over 90%.

(3) Optimization of the green FS information system

Green FS informatization refers to the realization of FS behavior by FS departments through computer communication and network technology. It can improve the speed of regulatory information exchange and promote information sharing. There is still great potential for the development of green financial regulatory information systems.

To understand the optimized performance of the green financial regulatory information system, various parameters were statistically analyzed with 50 data streams, Statistical information about the average speed of processing data streams includes the amount of data that can be processed per second and the distribution of processing time. Indicators such as average processing time, maximum processing time, and minimum processing time can be calculated to evaluate the responsiveness and efficiency of the system, as described in Table 3:

**Table 3.** Performance comparison of various parameters.

|  | Before Optimization | After Optimization |
|---|---|---|
| Aggregate throughput (Mbps) | 69.9 | 75.1 |
| Number of starving flows | 4 | 7 |
| Level of spatial reuse | 4.8 | 5.8 |
| Average transmission rate (Mbps) | 35 | 45 |

In Table 3, the optimized parameters significantly improved. The total throughput of the network increased from 69.9 (Mbps) to 75.1 (Mbps), and the average transmission rate also increased from 35 (Mbps) to 45 (Mpbs). This could indicate that the performance of the green financial regulatory information system optimized by GA significantly improved.

In order to analyze the application effectiveness of the green finance regulatory information system, this study took system users as the object and conducted a questionnaire evaluation survey on the usability and user satisfaction of the green finance regulatory information system before and after optimization. In the sample selection, to ensure the representativeness of the selected objects, this article randomly selected samples from renewable energy, clean technology, environmental protection, and other enterprises in different regions, industries, and scales of the Chinese market. This article divides the green finance market into different groups based on industry classification, and then randomly selects several groups for sampling. It conducts full member sampling in selected groups, which selects all individuals from each group as samples to ensure that the sample includes the survey subjects of each group, thereby better representing the overall characteristics of the green finance market. The main content of the questionnaire evaluation survey in this article includes comparative evaluation before and after use, evaluation of system functionality and performance, user experience and satisfaction, and other aspects. To ensure the independence of the survey, the questionnaire survey in this article was mainly conducted through online surveys, random interviews, and email surveys. The survey was conducted within a specific time period and adjusted flexibly according to the respondents' schedules. The first step was to provide a questionnaire to the respondents and collect their feedback. Then, the data were organized and statistically analyzed, and indicators such as frequency, percentage, and average were calculated for each problem. Finally, this article explains the evaluation results of the green finance regulatory information system before and after optimization based on the results and draws conclusions. The final results are shown in Figure 5.

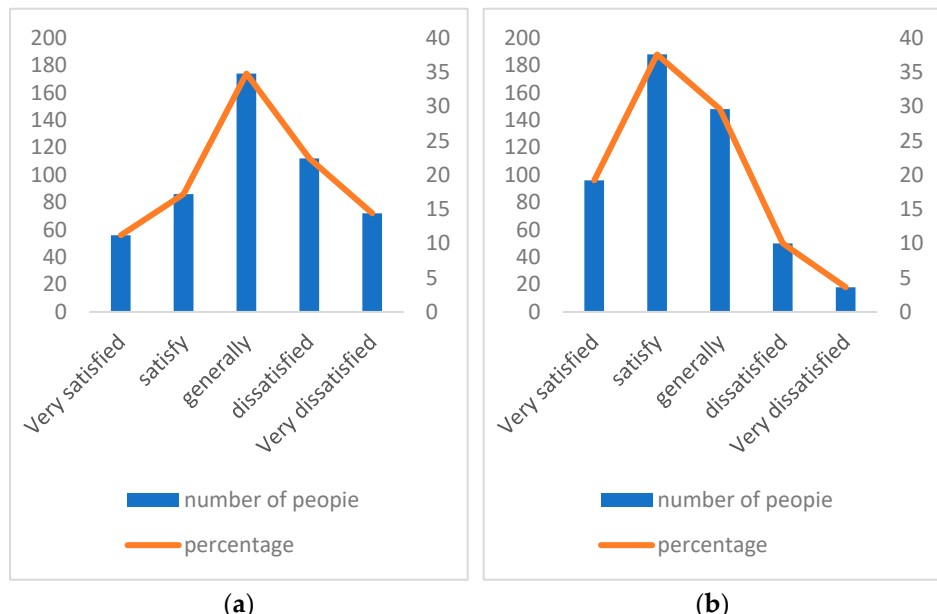

**Figure 5.** Investigation and evaluation of usage before and after system optimization. (**a**) Evaluation before system optimization; (**b**) evaluation after system optimization.

Figure 5 was prepared by the author. As shown in Figure 5, after conducting market research, the data showed that the proportion of people who were very satisfied with the optimized system increased from 11.2% to 19.2%. The proportion of people satisfied with it increased from 17.2% to 37.6%, and the number of people dissatisfied with the optimization significantly decreased. The proportion of people who were very dissatisfied with the system decreased from 14.4% to 3.6%. It could be seen that the usage of the optimized system was better.

The optimized GF supervision information system could better solve the current problems such as unstable system operation, unreasonable processes, weak statistical analysis capabilities, lack of business functions, and unfriendly interfaces. Through large-scale promotion and application and subsequent updates, it could effectively reduce human consumption in the GF supervision process, and improve work efficiency, so as to promote development. Through this optimization of the GF regulatory information system, more feasible suggestions for improving the GF regulatory information system in the future are proposed, and the shortcomings of this development are summarized in order to make timely adjustments for the future upgrade of the GF regulatory information system.

## 4. Discussion

To verify the effectiveness of the green finance regulatory information system optimized based on a genetic algorithm under carbon peak and carbon neutrality goals, this article analyzes the development and current status of green finance, as well as the optimization of the green finance regulatory information system from three perspectives:

(1) Green Finance Development Results

From the results of green finance development, compared to the national unit carbon emission GDP data, the unit carbon emission GDP of Region B also shows a continuous growth trend. The green finance regulatory information system has good analytical accuracy in the analysis of unit carbon emissions GDP in Region B and national unit carbon emissions GDP data. The green financial regulatory information system can ensure data consistency and standardization. Through a unified data collection and processing process, differences caused by different data sources and statistical methods can be avoided, and data consistency and comparability can be improved, thereby improving the accuracy of GDP data analysis.

(2)    Results of the current development status of green finance

From the current development status of green finance, it can be seen that the green finance market in China is showing a good development trend. From the data analysis of the green finance regulatory information system based on genetic algorithm optimization, the accuracy of China's carbon emissions GDP data and the balance of deposits and loans of various domestic financial institutions has reached over 90%. This indicates that the green finance regulatory information system has a great advantage in data analysis accuracy, providing more accurate, comprehensive, real-time, refined, and consistent green-finance-related data, thereby improving the objectivity of data analysis and evaluation.

## 5. Conclusions

The safe operation of the green financial regulatory information system relies on technical guarantees. Only when the regulatory technology and means are on the same platform as the operating technology of the regulated object can regulators ensure the effectiveness of supervision. To optimize the green financial regulatory information system, it is necessary to strengthen the exchange of information between various FIs within the jurisdiction, so as to improve the timeliness and practicality of information. Off-site supervision has become the mainstream of international FS, but the off-site supervision in China's regulatory methods is not strong enough. The monitoring indicator system also needs to be further improved, and the design of reports is not scientific enough. This paper used a GA to optimize the GF risk early warning and automatic early warning system in a carbon peaking and carbon neutrality environment, thus effectively promoting the effective supervision of GF. The research on green finance regulatory information systems based on genetic algorithm optimization is applicable to green finance regulation and decision-making under carbon peak and carbon neutrality goals. By optimizing algorithms, decision-makers can be assisted in formulating better green finance regulatory policies and measures under the constraints of carbon peak and carbon neutrality goals, thereby improving the efficiency and sustainability of the green finance market. Financial institutions and enterprises, as the main bodies of the green finance market, can understand the regulatory requirements and policy guidance of green finance through the regulatory information system under carbon peak and carbon neutrality goals, thereby adjusting their own green finance products and services and improving market competitiveness. Secondly, as regulators of the green financial market, government departments can systematically guide the formulation and implementation of relevant regulatory policies, promoting the development and standardization of the green financial market. Finally, investors and the public, as participants and beneficiaries of the green finance market, can understand the development trends and opportunities of the green finance market under carbon peak and carbon neutrality goals through system results, and make wiser investment decisions. However, the research process and this article still have some limitations. The optimization study of the green finance regulatory information system in this article only considers the constraints of carbon peak and carbon neutrality goals, but actual decision-making often involves balancing multiple factors, such as economic benefits and social sustainability. These factors may not be fully considered in the research, resulting in limitations in the research results. In future research, in order to promote the healthy development of the green financial market, the quality of research will be continuously improved from the perspective of influencing factors.

**Funding:** This research received no external funding.

**Institutional Review Board Statement:** Not applicable.

**Informed Consent Statement:** Not applicable.

**Data Availability Statement:** The data that support the findings of this study are available from the corresponding author upon reasonable request.

**Conflicts of Interest:** The authors declare no conflict of interest.

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
