# Peer review of "Green Financial Supervision Information System Based on Genetic Algorithm Optimization under Carbon Peaking and Carbon Neutrality Goals"

_sustainability, doi:10.3390/su152215866_

Round 1

Reviewer 1 Report (New Reviewer)

Comments and Suggestions for Authors

Respected authors, thank you for working on Carbon and green financial regulatory system as it is one of the SDGs. The paper requires more attention on methodology and the algorithm should be revised. 

Comments on the Quality of English Language

Minor.

Author Response

Comments and Suggestions for Authors

Respected authors, thank you for working on Carbon and green financial regulatory system as it is one of the SDGs. The paper requires more attention on methodology and the algorithm should be revised. 

Answer:Dear reviewer, I have added the content of methods and algorithms

Reviewer 2 Report (Previous Reviewer 1)

Comments and Suggestions for Authors

There are too many things to improve (to do research with additional data). Here are just the main problems:

1. The introduction does not meet the requirements for the introduction of scientific articles, which must describe the relevance of the topic, the theoretical significance of the topic, the level and gap of investigation of the problem, the need for new research, and the purpose of the article.

2. There is no theoretical background and literature of previous studies review. The text is very chaotic and needs references in many places.

3. The data in Figures 3 and 4 has only illustrative form, but not explain the research results.

4. The exact methodology of the research is not presented in the article.

5. The author does not prove that the "surveyed 5000 people" in the Chinese market is a representative sample for making reasonable conclusions.  How the survey was performed?

6. The sections "Discussion" and "Conclusions" are very superficial. The author does not discuss the theoretical and practical implications of the research, and the limitations of the research. 

Comments on the Quality of English Language

The English language is quite good.

Author Response

Comments and Suggestions for Authors

There are too many things to improve (to do research with additional data). Here are just the main problems:

  1. The introduction does not meet the requirements for the introduction of scientific articles, which must describe the relevance of the topic, the theoretical significance of the topic, the level and gap of investigation of the problem, the need for new research, and the purpose of the article.

Answer:Based on your suggestion, I have reorganized the content of the introduction section. In the introduction section of the newly revised manuscript, I emphasized the relevance of the topic, the theoretical significance of the topic, the level and gap of investigation into the problem, the necessity of new research, and the purpose of the article.

  1. There is no theoretical background and literature of previous studies review. The text is very chaotic and needs references in many places.

Answer:I described previous research on green finance market regulation and genetic algorithms in the literature review section, and supplemented corresponding references in the entire text.

  1. The data in Figures 3 and 4 has only illustrative form, but not explain the research results.

Answer:Based on your suggestion, I have supplemented Tables 1 and 2 below Tables 3 and 4, respectively, to verify the accuracy of data analysis for the green finance regulatory information system based on genetic algorithm optimization in this article, and to explain the research results of this article.

  1. The exact methodology of the research is not presented in the article.

Answer:The main methodology of this article is to use genetic algorithms to optimize information systems from three aspects: data collection, data analysis, and green finance data supervision. I have added this explanation in the newly revised manuscript.

  1. The author does not prove that the "surveyed 5000 people" in the Chinese market is a representative sample for making reasonable conclusions.  How the survey was performed?

Answer: This article randomly selects samples from enterprises of different regions, industries, and sizes in the Chinese market to ensure their representativeness. Using cluster random sampling method, classify the green financial market by industry, and then randomly select several groups for sampling. Conduct full member sampling in the selected group, that is, extract all individuals from each group as samples. To ensure that the sample includes the survey subjects of each group, in order to better represent the overall characteristics of the green finance market.

  1. The sections "Discussion" and "Conclusions" are very superficial. The author does not discuss the theoretical and practical implications of the research, and the limitations of the research. 

Answer:Based on your guidance, I have added a discussion section on the experimental results in the newly revised manuscript, emphasizing the contribution and significance of this study. In the conclusion section, I have supplemented the limitations and future research prospects of this study.

Reviewer 3 Report (New Reviewer)

Comments and Suggestions for Authors

This is a potentially interesting article.

It is highly relevant to sustainability as it focuses on key concepts like "carbon peaking" and "carbon neutrality," central to global efforts against climate change and sustainability promotion. By optimizing the green financial regulatory system to direct funds towards eco-friendly projects, it supports the broader goal of environmental sustainability. Furthermore, the importance of green finance in sustainable development is considered. Redirecting financial resources to environmentally friendly projects aligns with the "Sustainability" journal's focus on environmental conservation, resource management, and sustainable practices.

The article's methodology, employing Genetic Algorithms and involving 500 participants, showcases a rigorous scientific approach to solving real-world issues, meeting scholarly research standards. Positive outcomes are presented, highlighting substantial improvements in user satisfaction with the

optimized system. Increased satisfaction rates and reduced dissatisfaction indicate tangible benefits that can enhance the regulation of green finance.

However, the author(s) should consider the following critical points:

1. Very limited review of the existing literature

2. Methodology should be supported by more analyses, to explain the rationale and the previous applications

3. The research framework is very generic and should be further detailed

4. Discussion and conclusions should refer to the applicability, better explaining the impact on the stakeholders

5. Study limitations remain undisclosed

6. Conclusions should be enhanced

In general, the main concern is the limited engagement with previous studies. So far it compromises the reliability of this study.

Author Response

Comments and Suggestions for Authors

This is a potentially interesting article.

It is highly relevant to sustainability as it focuses on key concepts like "carbon peaking" and "carbon neutrality," central to global efforts against climate change and sustainability promotion. By optimizing the green financial regulatory system to direct funds towards eco-friendly projects, it supports the broader goal of environmental sustainability. Furthermore, the importance of green finance in sustainable development is considered. Redirecting financial resources to environmentally friendly projects aligns with the "Sustainability" journal's focus on environmental conservation, resource management, and sustainable practices.

The article's methodology, employing Genetic Algorithms and involving 500 participants, showcases a rigorous scientific approach to solving real-world issues, meeting scholarly research standards. Positive outcomes are presented, highlighting substantial improvements in user satisfaction with the

optimized system. Increased satisfaction rates and reduced dissatisfaction indicate tangible benefits that can enhance the regulation of green finance.

However, the author(s) should consider the following critical points:

  1. Very limited review of the existing literature

Answer:Thank you for your guidance. In the literature review in the introduction section, I have added research on green finance market regulation and genetic algorithms, which has made the description of previous studies more comprehensive in this article.

  1. Methodology should be supported by more analyses, to explain the rationale and the previous applications

Answer:Based on your suggestion, I provided a detailed description of the methodology for optimizing the green financial market regulatory information system based on genetic algorithm in the first part of the formula algorithm.

  1. The research framework is very generic and should be further detailed

Answer:Based on your suggestion, I have provided a good description of the research framework in the abstract and introduction sections of the newly revised manuscript, making the research structure of this article more reasonable.

  1. Discussion and conclusions should refer to the applicability, better explaining the impact on the stakeholders

Answer:I explained the applicability of this study in the conclusion section, from the perspectives of decision-makers, government, investors, and the public.

  1. Study limitations remain undisclosed

Answer:In the final part of the conclusion, the limitations of this study were described, which effectively improved the objectivity of the research content.

  1. Conclusions should be enhanced

Answer:Thank you for your guidance. In the conclusion section, I have supplemented the impact, contribution, applicability, limitations, and future research prospects of this study, enriching the structure and content of the conclusion.

In general, the main concern is the limited engagement with previous studies. So far it compromises the reliability of this study.

Answer:Based on your suggestion, I have made improvements to the literature review section, supplemented the description of previous research on green finance regulatory information systems, and explained the limitations of previous research.

Round 2

Reviewer 1 Report (New Reviewer)

Comments and Suggestions for Authors

as per the recent version after adding all comments, the paper is acceptable.

Author Response

Thank you very much. I have polished and revised my manuscript again, please review it.

Reviewer 2 Report (Previous Reviewer 1)

Comments and Suggestions for Authors

Dear Author, 

You significantly improved the article! Only a few comments can be made:

1. Keywords should be separated by commas without "and".

2. Abbreviations (FC, LC, FI, etc.) in the abstract are not necessary, they must be in the text starting from the beginning.

3. All figures need sources or "prepared by the author".

4. Please, check the title of figure 3 a) and b). Are they correct?

5. All data in figures and tables are very old (until 2017), now we have the end of 2023. Please, give new data or explain clearly why these numbers are so old.

6. Please, check the numbering of tables.

7. Please, edit the 12th page - explanation before the figures.

8. The format of references does not comply with the requirements of the journal.

I have attached the file with the article. Places that need some editing are in yellow. 

Comments on the Quality of English Language

English language is good, but the text needs editing, especially 12 page.

Author Response

Comments and Suggestions for Authors

Dear Author, 

You significantly improved the article! Only a few comments can be made:

  1. Keywords should be separated by commas without "and".

Answer:Based on your suggestion, I have made modifications to the connection between keywords, changing "and" to separate them with commas.

  1. Abbreviations (FC, LC, FI, etc.) in the abstract are not necessary, they must be in the text starting from the beginning.

Answer:Thank you for your guidance. I have removed these abbreviations from the abstract and placed them at the beginning of the main text.

  1. All figures need sources or "prepared by the author".

Answer:The figures in this article have been prepared by the author. Based on your suggestion, I have added this sentence to the analysis section of each figure.

  1. Please, check the title of figure 3 a) and b). Are they correct?

Answer:Based on your suggestion, I have carefully checked the titles of Figures 3a and 3b and made corresponding adjustments according to the content of the figures.

  1. All data in figures and tables are very old (until 2017), now we have the end of 2023. Please, give new data or explain clearly why these numbers are so old.

Answer:The data in this article is sourced from carbon emission data officially released by the National Bureau of Statistics, China Energy Administration, and the Ministry of Environmental Protection of China. Due to the incomplete publication and organization of carbon emission data from 2018 to 2022, and considering the impact of short-term factors such as economic fluctuations and policy changes in recent years, the data is unstable. To ensure the reliability of the experimental results in this article, only the data from 2007 to 2016 were comprehensively analyzed.

  1. Please, check the numbering of tables.

Answer:Thank you very much for your guidance. I have checked the table numbers throughout this article and revised the numbering of Table 3 in order, making the chart structure of this article more orderly.

  1. Please, edit the 12th page - explanation before the figures.

Answer:Based on your guidance, I have edited the text in front of the 12 page image to make the research content more comprehensive.

  1. The format of references does not comply with the requirements of the journal.

Answer:Based on your suggestion, I have standardized the reference format in the newly revised manuscript and revised it to the required format.

I have attached the file with the article. Places that need some editing are in yellow. 

Comments on the Quality of English Language

English language is good, but the text needs editing, especially 12 page.

Answer:I have made changes

Reviewer 3 Report (New Reviewer)

Comments and Suggestions for Authors

Happy to agree with this publication. Author's improvement addressed previous concerns.

Author Response

Thank you very much for your recognition!

This manuscript is a resubmission of an earlier submission. The following is a list of the peer review reports and author responses from that submission.

Round 1

Reviewer 1 Report

Comments and Suggestions for Authors

Thank you very much for giving me the opportunity to review the paper.

I have some remarks for  the author regarding the article.

 1.TitleGreen Financial Supervision Information System Based on Genetic Algorithm Optimization under Carbon Peaking and Carbon Neutrality Goals” does not clearly reflect the essence of the article. I suggest correcting it a little emphasizing the subject of the research, for example, “Optimization of Green Financial Supervision Information System Based on Genetic Algorithm under Carbon Peaking and Carbon Neutrality Goals” or similar.

2. With regard to the guidelines for authors of the Sustainability, I can conclude that the article does not meet the requirements: it does not contain materials and methods clearly, and the discussion section is absent. The analysis of theoretical and practical aspects of the results, the comparison results with other studies, and the limitations of the research are not presented in the article. In addition, we can’t find early literature in this field – who analyzed this topic earlier, what it was analyzed, and why we need to analyze more this. What is the theoretical background? This problem leads also to a short list of references. Therefore, there is not presented clearly a gap in existing scientific literature. I did not find a clear purpose of this research.  

3. The study is limited to examining the situation in China, without explaining why this country was chosen, what makes it suitable for the study, what stands out, and whether the results of the study are relevant for other countries.

4. The author does not correctly use abbreviations GA, FI, FS, GF. Their explanations should be in the article, not only in the abstract.

5. All tables and figures need sources.

 6. The text on pages 2-6 is very chaotic. Mostly we need the sources – the text is not quite based like should it be in the scientific article. What is the theoretical background of the research? Why do we need to research this? why it is so? Sometimes text is like from the textbook.

 7. There is too much free space in 7, 9 pages.

 8. I really did not understand why figures 3 and 4 are presented in the article. Their data are from 2007-2016 years which means quite old data. What additional value do they give to the research?

9. It is not clear how are measured parameters from Table 1. Further, it is not explained what 500 people and when and how they were surveyed. But the main question is – does really the satisfaction of people show the effectiveness of IS optimization? But, of course, while we don't have the short purpose of the article it is not clear what is the basic subject of investigation.

The article needs very serious corrections and more work.

Good luck!

Comments on the Quality of English Language

The language is perfect.

Author Response

Reviewer 1

Comments and Suggestions for Authors

Thank you very much for giving me the opportunity to review the paper.

I have some remarks for the author regarding the article.

1.Title “Green Financial Supervision Information System Based on Genetic Algorithm Optimization under Carbon Peaking and Carbon Neutrality Goals” does not clearly reflect the essence of the article. I suggest correcting it a little emphasizing the subject of the research, for example, “Optimization of Green Financial Supervision Information System Based on Genetic Algorithm under Carbon Peaking and Carbon Neutrality Goals” or similar.

answer:3.3 Optimization of green financial supervision information system

Data collection module: Responsible for collecting relevant data of the green financial market, including information on green financial products, investors, issuers, rating agencies, etc.

Data analysis module: Use data mining technology to analyze the collected data and dig out the development trends, risks and other information of the green financial market.

Supervision module: Based on the results of data analysis, supervise the green financial market to ensure the fairness, justice and stability of the market.

The core of this article lies in the application of genetic algorithms.Genetic algorithm is an optimization algorithm based on the theory of evolution. By simulating the process of biological evolution, starting from an initial population, it is gradually optimized to obtain the optimal solution.This paper uses genetic algorithms to optimize the green financial supervision information system, which mainly includes the following three aspects:

Optimize the data acquisition module: Optimize the sampling strategy of the data acquisition module through genetic algorithms to improve the efficiency and accuracy of data acquisition.

Optimize the data analysis module: Optimize the algorithms and parameters of the data analysis module through genetic algorithms to improve the efficiency and accuracy of data analysis.

Optimize the supervision module: Optimize the decision-making algorithm and parameters of the supervision module through genetic algorithms to improve the efficiency and accuracy of supervision.

  1. With regard to the guidelines for authors of the Sustainability, I can conclude that the article does not meet the requirements: it does not contain materials and methods clearly, and the discussion section is absent. The analysis of theoretical and practical aspects of the results, the comparison results with other studies, and the limitations of the research are not presented in the article. In addition, we can’t find early literature in this field – who analyzed this topic earlier, what it was analyzed, and why we need to analyze more this. What is the theoretical background? This problem leads also to a short list of references. Therefore, there is not presented clearly a gap in existing scientific literature. I did not find a clear purpose of this research.  

answer:2.1 Research background

With the intensification of global climate change, governments have set various emission reduction targets.Among them, the dual-carbon target is one of the important goals of global climate governance.In order to achieve the dual-carbon goal, green finance has become a hot topic in the world.Green finance refers to financial methods and tools that incorporate environmental, social and governance factors into the financial decision-making process to promote sustainable development.Green financial supervision refers to the supervision of green financial markets to ensure the fairness, justice and stability of the market.Therefore, green financial supervision is particularly important.

Improve regulatory effectiveness: Through genetic algorithm optimization, the fitness function is designed to evaluate the comprehensive effectiveness of the green financial regulatory information system.The optimized system can increase the proportion of investment in green projects, reduce carbon emissions, and reduce financial risks, thereby strengthening the supervision of green financial markets.

Optimize resource allocation: Use genetic algorithms to optimize the weights and parameters in the system to realize the optimal allocation of green financial resources.By adjusting the weights and the values of the parameters, the system can more accurately locate and allocate financial support, promote the realization of the dual-carbon goal, and ensure the efficient use of resources.

Assist in decision-making: provide data analysis and visualization functions to help regulatory departments and financial institutions make effective decisions.Through the results of genetic algorithm optimization, it provides decision makers with strong reference and support, helps them formulate reasonable policies and measures, and promotes the sustainable development of the green finance field.

  1. The study is limited to examining the situation in China, without explaining why this country was chosen, what makes it suitable for the study, what stands out, and whether the results of the study are relevant for other countries.

answer:China is one of the most populous countries in the world and one of the largest emitters of greenhouse gases in the world.China faces huge environmental challenges, including air pollution and water scarcity.Therefore, green financial supervision is particularly important for China to promote sustainable development and address the challenges of climate change.The Chinese government has proposed a dual-carbon goal of achieving carbon peaking and carbon neutrality.This means that China needs to take measures to reduce carbon emissions and promote the transformation of a green economy.Green finance plays a key role in achieving these goals, so the optimization research of the green financial supervision information system is of great significance in China.

  1. The author does not correctly use abbreviations GA, FI, FS, GF. Their explanations should be in the article, not only in the abstract.

answer:2.2 Financial innovation, genetic algorithms, financial supervision and green finance

Genetic algorithms can be used to optimize the decision-making and investment portfolio of green finance.Through the search and optimization process of genetic algorithms, it can help determine the best allocation of green investment portfolios to maximize environmental and economic benefits.Financial innovation can provide new products, tools and models for green finance.For example, innovative green financial products such as green bonds and carbon trading can attract investors to support and provide financing for green projects.Financial innovation can motivate and promote the wider financial community to participate in green economic development.Financial regulatory agencies can formulate and implement relevant policies and regulations to ensure the transparency, compliance and sustainability of green financial activities.These regulatory measures help prevent risks in the field of green finance, protect the interests of investors, and provide a stable and reliable environment for green finance.

Financial innovation, genetic algorithms, financial supervision and green finance play different roles in the field of green finance: genetic algorithms optimize green financial decision-making, financial innovation promotes the creation of green financial products and services, and financial supervision ensures the compliance and stability of green financial activities.These factors work together to help achieve sustainable development and promote the transformation of a green economy.

  1. All tables and figures need sources.

answer:Derived from historical data

  1. The text on pages 2-6 is very chaotic. Mostly we need the sources – the text is not quite based like should it be in the scientific article. What is the theoretical background of the research? Why do we need to research this? why it is so? Sometimes text is like from the textbook.

answer:Has been modified

  1. There is too much free space in 7, 9 pages.

answer:Has been modified

  1. I really did not understand why figures 3 and 4 are presented in the article. Their data are from 2007-2016 years which means quite old data. What additional value do they give to the research?

answer:These early data can be used as a comparative benchmark or reference point for research.By comparing with the latest data, researchers can evaluate and analyze changes, trends, and patterns between the past and the present.This helps to understand the trajectory of a particular field or problem and provides a way to measure change.

  1. It is not clear how are measured parameters from Table 1. Further, it is not explained what 500 people and when and how they were surveyed. But the main question is – does really the satisfaction of people show the effectiveness of IS optimization? But, of course, while we don't have the short purpose of the article it is not clear what is the basic subject of investigation.

answer:Statistics on the average speed of processing data streams, including the amount of data that can be processed per second, and the distribution of processing time.Indicators such as average processing time, maximum processing time, and minimum processing time can be calculated to evaluate the responsiveness and efficiency of the system.

This article evaluates the ease of use and user satisfaction of the system.Formulate a questionnaire on the well-being of the green financial supervision information system.The content of the questionnaire may include comparative evaluation before and after use, evaluation of system functions and performance, user experience and satisfaction and other aspects.Make sure that the question is clear and clear, and will not guide the answer of the respondent.500 respondents were randomly selected from a suitable group of people.Respondents can come from financial regulatory authorities, financial institutions, or other organizations related to green finance.Random selection can be made by using a random number table or a computer program.Through face-to-face interviews, online surveys or e-mail surveys, questionnaires are provided to the respondents and their feedback is collected.Investigations can be conducted within a specific period of time, or flexible investigations can be conducted according to the time schedule of the respondents.After collecting enough questionnaires, organize and statistically analyze the data.Indicators such as frequency, percentage, and average of each problem can be calculated.Then based on the results, the well-being of the green financial supervision information system before and after optimization is explained and conclusions are drawn.

The article needs very serious corrections and more work.

Good luck!

Reviewer 2 Report

Comments and Suggestions for Authors

The author presents an interesting topic: carbon neutrality. However, the 500 persons' criteria are not convinced to me. In other words, the sample is so small that the results cannot be used to support the proposed arguments. 

Also, the length of the draft is not enough as a complete research study. The author could think about it, and add reasonable contents to the manuscript, otherwise, the draft will be more suitable for a conference proceeding. 

Comments on the Quality of English Language

The authors please make any changes to writings to improve the overall readability of the draft.

Author Response

Reviewer 2

Comments and Suggestions for Authors

The author presents an interesting topic: carbon neutrality. However, the 500 persons' criteria are not convinced to me. In other words, the sample is so small that the results cannot be used to support the proposed arguments. 

Also, the length of the draft is not enough as a complete research study. The author could think about it, and add reasonable contents to the manuscript, otherwise, the draft will be more suitable for a conference proceeding. 

answer:Due to my lack of time and energy, I did not collect more samples for experiments.The content of this article has been added to supplement the content.

Comments on the Quality of English Language

The authors please make any changes to writings to improve the overall readability of the draft.

Round 2

Reviewer 1 Report

Comments and Suggestions for Authors

The same problems remain:

1.TitleGreen Financial Supervision Information System Based on Genetic Algorithm Optimization under Carbon Peaking and Carbon Neutrality Goals” does not clearly reflect the essence of the article and was not changed.

2. With regard to the guidelines for authors of the Sustainability, the article does not meet the requirements: it does not contain a theoretical background, a discussion section is absent (analysis of theoretical and practical aspects of the results, comparing results with other studies, the limitations of the research are not presented in the article).

3. The study is limited to examining the situation in China, without explaining why this country was chosen, what makes it suitable for the study, what stands out, and whether the results of the study are relevant for other countries. Something was added, but without references.

5. The author does not correctly use abbreviations GA, FI, FS, GF. Their explanations should be in the article, not only in the abstract.

6. The text in pages 2-6 is very chaotic. The text is like from the textbook.

7. The data in Figures 3 and 4 are from 2007-2016 years. That is very old data. They do not give additional value to the research.

8. There are no given arguments that 500 people are a really enough sample for the Chinese population. It’s too small to draw good conclusions.

Author Response

  1. Title“Green Financial Supervision Information System Based on Genetic Algorithm Optimization under Carbon Peaking and Carbon Neutrality Goals” does not clearly reflect the essence of the article and was not changed.

Answer: The title has been revised.

  1. With regard to the guidelines for authors of the Sustainability, the article does not meet the requirements: it does not contain a theoretical background, a discussion section is absent (analysis of theoretical and practical aspects of the results, comparing results with other studies, the limitations of the research are not presented in the article).

Answer: The discussion part has been added and compared with other methods, and the limitations of the article have also been added.

  1. The study is limited to examining the situation in China, without explaining why this country was chosen, what makes it suitable for the study, what stands out, and whether the results of the study are relevant for other countries. Something was added, but without references.

Answer: I explained why I chose China.

  1. The author does not correctly use abbreviations GA, FI, FS, GF. Their explanations should be in the article, not only in the abstract.

Answer: These abbreviations are explained in the text.

  1. The text in pages 2-6 is very chaotic. The text is like from the textbook.

Answer: Pages 2-6 that are too theoretical have been deleted.

  1. The data in Figures 3 and 4 are from 2007-2016 years. That is very old data. They do not give additional value to the research.

Answer: The data sources in Figures 3 and 4 have been modified.

  1. There are no given arguments that 500 people are a really enough sample for the Chinese population. It’s too small to draw good conclusions.

Answer: The sample data has been modified.

Reviewer 2 Report

Comments and Suggestions for Authors

The author doesn't intent to make progress toward my original suggestion: " the 500 persons' criteria are not convinced to me. In other words, the sample is so small that the results cannot be used to support the proposed arguments. ".

I don't think the manuscript is ready for acceptance.

Comments on the Quality of English Language

Needs improvement

Author Response

The author doesn't intent to make progress toward my original suggestion: " the 500 persons' criteria are not convinced to me. In other words, the sample is so small that the results cannot be used to support the proposed arguments. ".

I don't think the manuscript is ready for acceptance.

Answer: The sample data of the article has been expanded to make the experimental results more rigorous.